# Learning Disentangled Joint Continuous and Discrete Representations

**Emilien Dupont**
Schlumberger Software Technology Innovation Center
Menlo Park, CA, USA
dupont@slb.com

## Abstract

We present a framework for learning disentangled and interpretable jointly continuous and discrete representations in an unsupervised manner. By augmenting the continuous latent distribution of variational autoencoders with a relaxed discrete distribution and controlling the amount of information encoded in each latent unit, we show how continuous and categorical factors of variation can be discovered automatically from data. Experiments show that the framework disentangles continuous and discrete generative factors on various datasets and outperforms current disentangling methods when a discrete generative factor is prominent.

## 1 Introduction

Disentangled representations are defined as ones where a change in a single unit of the representation corresponds to a change in single factor of variation of the data while being invariant to others (Bengio et al. (2013)). For example, a disentangled representation of 3D objects could contain a set of units each corresponding to a distinct generative factor such as position, color or scale. Most recent work on learning disentangled representations has focused on modeling continuous factors of variation (Higgins et al. (2016); Kim & Mnih (2018); Chen et al. (2018)). However, a large number of datasets contain inherently discrete generative factors which can be difficult to capture with these methods. In image data for example, distinct objects or entities would most naturally be represented by discrete variables, while their position or scale might be represented by continuous variables.

Several machine learning tasks, including transfer learning and zero-shot learning, can benefit from disentangled representations (Lake et al. (2017)). Disentangled representations have also been applied to reinforcement learning (Higgins et al. (2017a)) and for learning visual concepts (Higgins et al. (2017b)). Further, in contrast to most representation learning algorithms, disentangled representations are often interpretable since they align with factors of variation of the data. Different approaches have been explored for semi-supervised or supervised learning of factored representations (Kulkarni et al. (2015); Whitney et al. (2016); Yang et al. (2015); Reed et al. (2014)). These approaches achieve impressive results but either require knowledge of the underlying generative factors or other forms of weak supervision. Several methods also exist for unsupervised disentanglement with the two most prominent being InfoGAN and $\beta$-VAE (Chen et al. (2016); Higgins et al. (2016)). These frameworks have shown promise in disentangling factors of variation in an unsupervised manner on a number of datasets.

InfoGAN (Chen et al. (2016)) is a framework based on Generative Adversarial Networks (Goodfellow et al. (2014)) which disentangles generative factors by maximizing the mutual information between a subset of latent variables and the generated samples. While this approach is able to model both discrete and continuous factors, it suffers from some of the shortcomings of Generative Adversarial Networks (GAN), such as unstable training and reduced sample diversity. Recent improvements in the training of GANs (Arjovsky et al. (2017); Gulrajani et al. (2017)) have mitigated some of these

issues, but stable GAN training still remains a challenge (and this is particularly challenging for InfoGAN as shown in Kim & Mnih (2018)). $\beta$-VAE (Higgins et al. (2016)), in contrast, is based on Variational Autoencoders (Kingma & Welling (2013); Rezende et al. (2014)) and is stable to train. $\beta$-VAE, however, can only model continuous latent variables.

In this paper we propose a framework, based on Variational Autoencoders (VAE), that learns disentangled continuous and discrete representations in an unsupervised manner. It comes with the advantages of VAEs, such as stable training, large sample diversity and a principled inference network, while having the flexibility to model a combination of continuous and discrete generative factors. We show how our framework, which we term JointVAE, discovers independent factors of variation on MNIST, FashionMNIST (Xiao et al. (2017)), CelebA (Liu et al. (2015)) and Chairs (Aubry et al. (2014)). For example, on MNIST, JointVAE disentangles digit type (discrete) from slant, width and stroke thickness (continuous). In addition, the model's learned inference network can infer various properties of data, such as the azimuth of a chair, in an unsupervised manner. The model can also be used for simple image editing, such as rotating a face in an image.

## 2    Analysis of $\beta$-VAE

We derive our approach by modifying the $\beta$-VAE framework and augmenting it with a joint latent distribution. $\beta$-VAEs model a joint distribution of the data $\mathbf{x}$ and a set of latent variables $\mathbf{z}$ and learn continuous disentangled representations by maximizing the objective

$$\mathcal{L}(\theta, \phi) = \mathbb{E}_{q_\phi(\mathbf{z}|\mathbf{x})}[\log p_\theta(\mathbf{x}|\mathbf{z})] - \beta D_{KL}(q_\phi(\mathbf{z}|\mathbf{x}) \parallel p(\mathbf{z})) \tag{1}$$

where the posterior or encoder $q_\phi(\mathbf{z}|\mathbf{x})$ is a neural network with parameters $\phi$ mapping $\mathbf{x}$ into $\mathbf{z}$, the likelihood or decoder $p_\theta(\mathbf{x}|\mathbf{z})$ is a neural network with parameters $\theta$ mapping $\mathbf{z}$ into $\mathbf{x}$ and $\beta$ is a positive constant. The loss is a weighted sum of a likelihood term $\mathbb{E}_{q_\phi(\mathbf{z}|\mathbf{x})}[\log p_\theta(\mathbf{x}|\mathbf{z})]$ which encourages the model to encode the data $\mathbf{x}$ into a set of latent variables $\mathbf{z}$ which can efficiently reconstruct the data and a second term that encourages the distribution of the inferred latents $\mathbf{z}$ to be close to some prior $p(\mathbf{z})$. When $\beta = 1$, this corresponds to the original VAE framework. However, when $\beta > 1$, it is theorized that the increased pressure of the posterior $q_\phi(\mathbf{z}|\mathbf{x})$ to match the prior $p(\mathbf{z})$, combined with maximizing the likelihood term, gives rises to efficient and disentangled representations of the data (Higgins et al. (2016); Burgess et al. (2017)).

We can derive further insights by analyzing the role of the KL divergence term in the objective (1). During training, the objective will be optimized in expectation over the data $\mathbf{x}$. The KL term then becomes (Makhzani & Frey (2017); Kim & Mnih (2018))

$$\mathbb{E}_{p(\mathbf{x})}[D_{KL}(q_\phi(\mathbf{z}|\mathbf{x}) \parallel p(\mathbf{z}))] = I(\mathbf{x}; \mathbf{z}) + D_{KL}(q(\mathbf{z}) \parallel p(\mathbf{z}))$$
$$\geq I(\mathbf{x}; \mathbf{z}) \tag{2}$$

i.e., when taken in expectation over the data, the KL divergence term is an upper bound on the mutual information between the latents and the data (see appendix for proof and details). Thus, a mini batch estimate of the mean KL divergence is an estimate of the upper bound on the information $\mathbf{z}$ can transmit about $\mathbf{x}$.

Penalizing the mutual information term improves disentanglement but comes at the cost of increased reconstruction error. Recently, several methods have been explored to improve the reconstruction quality without decreasing disentanglement (Burgess et al. (2017); Kim & Mnih (2018); Chen et al. (2018); Gao et al. (2018)). Burgess et al. (2017) in particular propose an objective where the upper bound on the mutual information is controlled and gradually increased during training. Denoting the controlled information capacity by $C$, the objective is defined as

$$\mathcal{L}(\theta, \phi) = \mathbb{E}_{q_\phi(\mathbf{z}|\mathbf{x})}[\log p_\theta(\mathbf{x}|\mathbf{z})] - \gamma |D_{KL}(q_\phi(\mathbf{z}|\mathbf{x}) \parallel p(\mathbf{z})) - C| \tag{3}$$

where $\gamma$ is a constant which forces the KL divergence term to match the capacity $C$. Gradually increasing $C$ during training allows for control of the amount of information the model can encode. This objective has been shown to improve reconstruction quality as compared to (1) without reducing disentanglement (Burgess et al. (2017)).

# 3 JointVAE Model

We propose a modification to the $\beta$-VAE framework which allows us to model a joint distribution of continuous and discrete latent variables. Letting $\mathbf{z}$ denote a set of continuous latent variables and $\mathbf{c}$ denote a set of categorical or discrete latent variables, we define a joint posterior $q_\phi(\mathbf{z}, \mathbf{c}|\mathbf{x})$, prior $p(\mathbf{z}, \mathbf{c})$ and likelihood $p_\theta(\mathbf{x}|\mathbf{z}, \mathbf{c})$. The $\beta$-VAE objective then becomes

$$\mathcal{L}(\theta, \phi) = \mathbb{E}_{q_\phi(\mathbf{z}, \mathbf{c}|\mathbf{x})}[\log p_\theta(\mathbf{x}|\mathbf{z}, \mathbf{c})] - \beta D_{KL}(q_\phi(\mathbf{z}, \mathbf{c}|\mathbf{x}) \,\|\, p(\mathbf{z}, \mathbf{c})) \tag{4}$$

where the latent distribution is now jointly continuous and discrete. Assuming the continuous and discrete latent variables are conditionally independent[1], i.e. $q_\phi(\mathbf{z}, \mathbf{c}|\mathbf{x}) = q_\phi(\mathbf{z}|\mathbf{x})q_\phi(\mathbf{c}|\mathbf{x})$ and similarly for the prior $p(\mathbf{z}, \mathbf{c}) = p(\mathbf{z})p(\mathbf{c})$ we can rewrite the KL divergence as

$$D_{KL}(q_\phi(\mathbf{z}, \mathbf{c}|\mathbf{x}) \,\|\, p(\mathbf{z}, \mathbf{c})) = D_{KL}(q_\phi(\mathbf{z}|\mathbf{x}) \,\|\, p(\mathbf{z})) + D_{KL}(q_\phi(\mathbf{c}|\mathbf{x}) \,\|\, p(\mathbf{c})) \tag{5}$$

i.e. we can separate the discrete and continuous KL divergence terms (see appendix for proof). Under this assumption, the loss becomes

$$\mathcal{L}(\theta, \phi) = \mathbb{E}_{q_\phi(\mathbf{z}, \mathbf{c}|\mathbf{x})}[\log p_\theta(\mathbf{x}|\mathbf{z}, \mathbf{c})] - \beta D_{KL}(q_\phi(\mathbf{z}|\mathbf{x}) \,\|\, p(\mathbf{z})) - \beta D_{KL}(q_\phi(\mathbf{c}|\mathbf{x}) \,\|\, p(\mathbf{c})) \tag{6}$$

In our initial experiments, we found that directly optimizing this loss led to the model ignoring the discrete latent variables. Similarly, gradually increasing the channel capacity as in equation (3) leads to the model assigning all capacity to the continuous channels. To overcome this, we split the capacity increase: the capacities of the discrete and continuous latent channels are controlled separately forcing the model to encode information both in the discrete and continuous channels. The final loss is then given by

$$\mathcal{L}(\theta, \phi) = \mathbb{E}_{q_\phi(\mathbf{z}, \mathbf{c}|\mathbf{x})}[\log p_\theta(\mathbf{x}|\mathbf{z}, \mathbf{c})] - \gamma|D_{KL}(q_\phi(\mathbf{z}|\mathbf{x}) \,\|\, p(\mathbf{z})) - C_z| - \gamma|D_{KL}(q_\phi(\mathbf{c}|\mathbf{x}) \,\|\, p(\mathbf{c})) - C_c| \tag{7}$$

where $C_z$ and $C_c$ are gradually increased during training.

## 3.1 Parametrization of continuous latent variables

As in the original VAE framework, we parametrize $q_\phi(\mathbf{z}|\mathbf{x})$ by a factorised Gaussian, i.e. $q_\phi(\mathbf{z}|\mathbf{x}) = \prod_i q_\phi(z_i|\mathbf{x})$ where $q_\phi(z_i|\mathbf{x}) = \mathcal{N}(\mu_i, \sigma_i^2)$ and let the prior be a unit Gaussian $p(\mathbf{z}) = \mathcal{N}(0, I)$. $\boldsymbol{\mu}$ and $\boldsymbol{\sigma^2}$ are both parametrized by neural networks.

## 3.2 Parametrization of discrete latent variables

Parametrizing $q_\phi(\mathbf{c}|\mathbf{x})$ is more difficult. Since $q_\phi(\mathbf{c}|\mathbf{x})$ needs to be differentiable with respect to its parameters, we cannot parametrize $q_\phi(\mathbf{c}|\mathbf{x})$ by a set of categorical distributions. Recently, Maddison et al. (2016) and Jang et al. (2016) proposed a differentiable relaxation of discrete random variables based on the Gumbel Max trick (Gumbel (1954)). If $c$ is a categorical variable with class probabilities $\alpha_1, \alpha_2, ..., \alpha_n$, then we can sample from a continuous approximation of the categorical distribution, by sampling a set of $g_k \sim \text{Gumbel}(0, 1)$ i.i.d. and applying the following transformation

$$y_k = \frac{\exp((\log \alpha_k + g_k)/\tau)}{\sum_i \exp((\log \alpha_i + g_i)/\tau)} \tag{8}$$

where $\tau$ is a temperature parameter which controls the relaxation. The sample $\mathbf{y}$ is a continuous approximation of the one hot representation of $\mathbf{c}$. The relaxed discrete distribution is called a Concrete or Gumbel Softmax distribution and is denoted by $g(\boldsymbol{\alpha})$ where $\boldsymbol{\alpha}$ is a vector of class probabilities.

We can parametrize $q_\phi(\mathbf{c}|\mathbf{x})$ by a product of independent Gumbel Softmax distributions, $q_\phi(\mathbf{c}|\mathbf{x}) = \prod_i q_\phi(c_i|\mathbf{x})$ where $q_\phi(c_i|\mathbf{x}) = g(\boldsymbol{\alpha}^{(i)})$ is a Gumbel Softmax distribution with class probabilities $\boldsymbol{\alpha}^{(i)}$. We let the prior $p(\mathbf{c})$ be equal to a product of uniform Gumbel Softmax distributions. This approach allows us to use the reparametrization trick (Kingma & Welling (2013); Rezende et al. (2014)) and efficiently train the discrete model.

### 3.3 Architecture

The final architecture of the JointVAE model is shown in Fig. 1. We build the encoder to output the parameters of the continuous distribution $\boldsymbol{\mu}$ and $\boldsymbol{\sigma^2}$ and of each of the discrete distributions $\boldsymbol{\alpha}^{(i)}$. We then sample $z_i \sim \mathcal{N}(\mu_i, \sigma_i^2)$ and $c_i \sim g(\boldsymbol{\alpha}^{(i)})$ using the reparametrization trick and concatenate $\mathbf{z}$ and $\mathbf{c}$ into one latent vector which is passed as input to the decoder.

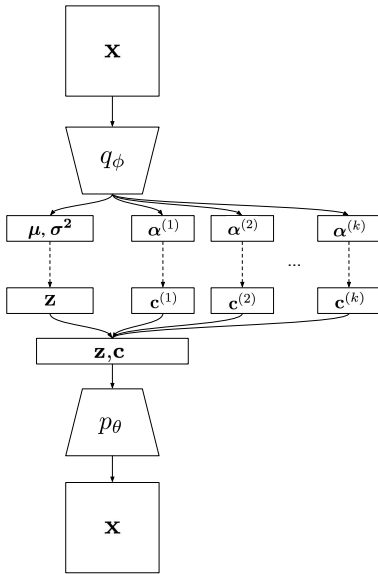

Figure 1: JointVAE architecture. The input $\mathbf{x}$ is encoded by $q_\phi$ into the parameters of the latent distributions. Samples are drawn from each of the latent distributions using the reparametrization trick (indicated by dashed arrows on the diagram). The samples are then concatenated and decoded through $p_\theta$.

### 3.4 Choice and sensitivity hyperparameters

The JointVAE loss in equation 7 depends on the hyperparameters $\gamma$, $C_c$ and $C_z$. While the choice of these is ultimately empirical, there are various heuristics we can use to narrow the search. The value of $\gamma$, for example, is chosen so that it is large enough to maintain the capacity at the desired level (e.g. large improvements in reconstruction error should not come at the cost of breaking the capacity constraint). We found the model to be quite robust to changes in $\gamma$. As the capacity of a discrete channel is bounded, $C_c$ is chosen to be the maximum capacity of the channel, encouraging the model to use all categories of the discrete distribution. $C_z$ is more difficult to choose and is often chosen by experiment to be the largest value where the representation is still disentangled (in a similar way that $\beta$ is chosen as the lowest value where the representation is still disentangled in $\beta$-VAE).

## 4 Experiments

We perform experiments on several datasets including MNIST, FashionMNIST, CelebA and Chairs. We parametrize the encoder by a convolutional neural network and the decoder by the same network, transposed (for the full architecture and training details see appendix). The code, along with all experiments and trained models presented in this paper, is available at `https://github.com/Schlumberger/joint-vae`.

**MNIST**

Disentanglement results and latent traversals for MNIST are shown in Fig. 2. The model was trained with 10 continuous latent variables and one discrete 10-dimensional latent variable. The model discovers several factors of variation in the data, such as digit type (discrete), stroke thickness, angle and width (continuous) in an unsupervised manner. As can be seen from the latent traversals in Fig. 2, the trained model is able to generate realistic samples for a large variety of latent settings. Fig. 4a shows digits generated by fixing the discrete latent and sampling the continuous latents from the prior $p(\mathbf{z}) = \mathcal{N}(0, 1)$, which can be interpreted as sampling from a distribution conditioned on digit type. As can be seen, the samples are diverse, realistic and honor the conditioning.

For a large range of hyperparameters we were not able to achieve disentanglement using the purely continuous $\beta$-VAE framework (see Fig. 3). This is likely because MNIST has an inherently discrete generative factor (digit type), which $\beta$-VAE is unable to map onto a continuous latent variable. In contrast, the JointVAE approach allows us to disentangle the discrete factors while maintaining disentanglement of continuous factors. To the best of our knowledge, JointVAE is, apart from InfoGAN, the only framework which disentangles MNIST in a completely unsupervised manner and it does so in a more stable way than InfoGAN.

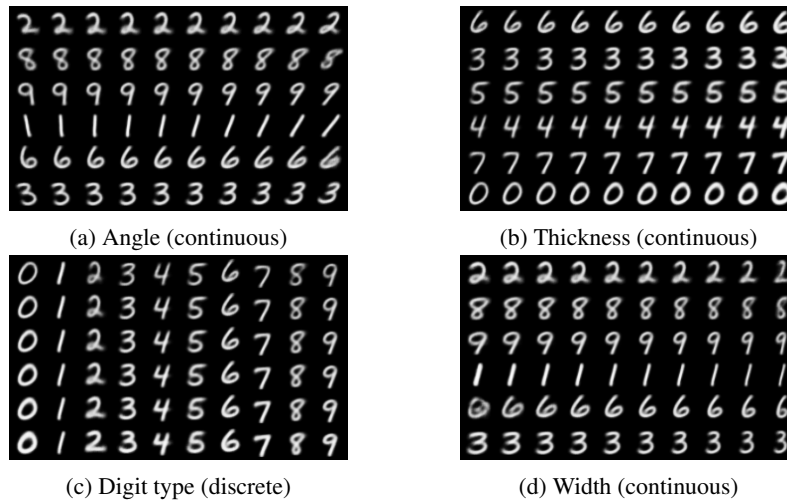

(a) Angle (continuous)          (b) Thickness (continuous)

(c) Digit type (discrete)          (d) Width (continuous)

Figure 2: Latent traversals of the model trained on MNIST with 10 continuous latent variables and 1 discrete latent variable. Each row corresponds to a fixed random setting of the latent variables and the columns correspond to varying a single latent unit. Each subfigure varies a different latent unit. As can be seen each of the varied latent units corresponds to an interpretable generative factor, such as stroke thickness or digit type.

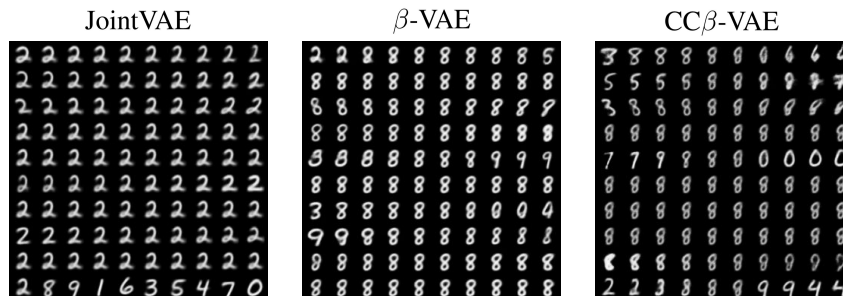

Figure 3: Traversals of all latent dimensions on MNIST for JointVAE, $\beta$-VAE and $\beta$-VAE with controlled capacity increase (CC$\beta$-VAE). JointVAE is able to disentangle digit type from continuous factors of variation like stroke thickness and angle, while digit type is entangled with continuous factors for both $\beta$-VAE and CC$\beta$-VAE.

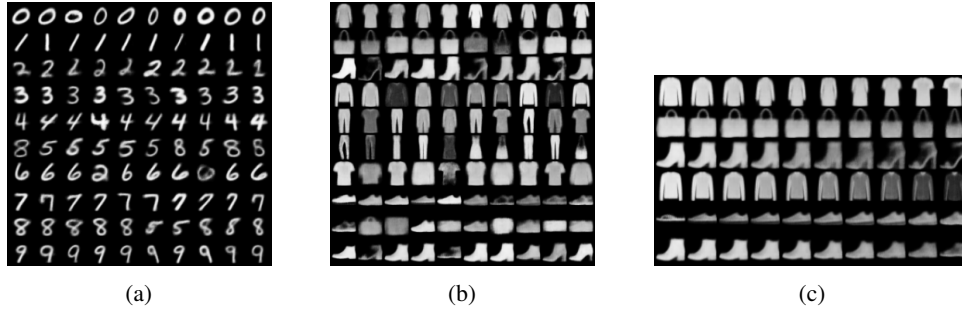

| (a) | (b) | (c) |

Figure 4: (a) Samples conditioned on digit type. Each row shows samples from $p_\theta$ where the discrete latent variable is fixed and all other latent values are sampled from the prior. As can be seen each row produces diverse samples of each digit. Note that digits which are similar, such as 5 and 8 are sometimes confused and not perfectly disentangled. (b) Samples conditioned on fashion item type. The samples are diverse and largely disentangled. (c) Latent traversals of FashionMNIST model. The rows correspond to different settings of the discrete latent variable, while the columns correspond to a traversal of the most informative continuous latent variable. Various factors are discovered, such as sleeve length, bag handle size, ankle height and shoe opening.

**FashionMNIST**

Latent traversals for FashionMNIST are shown in Fig. 4c. We also used 10 continuous and 1 discrete latent variable for this dataset. FashionMNIST is harder to disentangle as the generative factors for creating clothes are not as clear as the ones for drawing digits. However, JointVAE performs well and discovers interesting dimensions, such as sleeve length, heel size and shirt color. As some of the classes of FashionMNIST are very similar (e.g. shirt and t-shirt are two different classes), not all classes are discovered. However, a significant amount of them are disentangled including dress, t-shirt, trousers, sneakers, bag, ankle boot and so on (see Fig. 4b).

**CelebA**

For CelebA we used a model with 32 continuous latent variables and one 10 dimensional discrete latent variable. As shown in Fig. 5, the JointVAE model discovers various factors of variation including azimuth, age and background color, while being able to generate realistic samples. Different settings of the discrete variable correspond to different facial identities. While the samples are not as sharp as those produced by entangled models, we can still see details in the images such as distinct facial features and skin tones (the trade-off between disentanglement and reconstruction quality is a well known problem which is discussed in Higgins et al. (2016); Burgess et al. (2017); Kim & Mnih (2018); Chen et al. (2018)).

**Chairs**

For the chairs dataset we used a model with 32 continuous latent variables and 3 binary discrete latent variables. JointVAE discovers several factors of variation such as chair rotation, width and leg style. Furthermore, different settings of the discrete variables correspond to different chair types and colors.

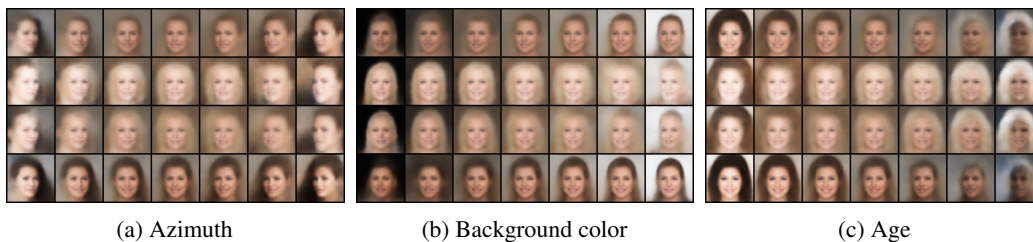

| (a) Azimuth | (b) Background color | (c) Age |

Figure 5: Latent traversals of the model trained on CelebA. Each row corresponds to a fixed setting of the discrete latent variable and the columns correspond to varying a single continuous latent unit.

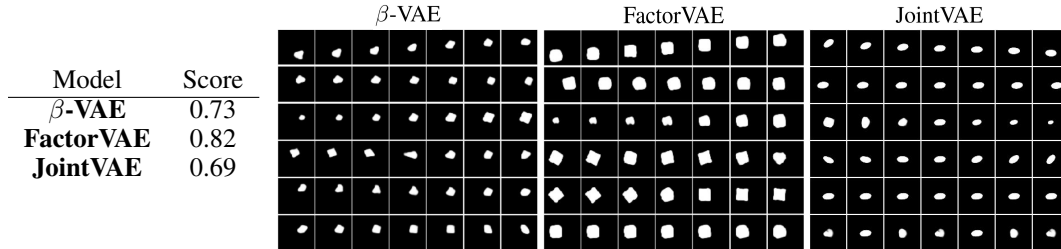

| Model | Score |
|---|---|
| $\beta$-**VAE** | 0.73 |
| **FactorVAE** | 0.82 |
| **JointVAE** | 0.69 |

Figure 6: *Left*: Disentanglement scores for various frameworks on the dSprites dataset. The scores are obtained by averaging scores over 10 different random seeds from the model with the best hyperparameters (removing outliers where the model collapsed to the mean). *Right*: Comparison of latent traversals on the dSprites dataset. There are 4 continuous factors and 1 discrete factor in the original dataset and only JointVAE is able to encode all information into 4 continuous and 1 discrete latent variables. Note that the final row of the JointVAE latent traversal corresponds to the discrete factor of dimension 3, which is why the patterns repeat with a period of 3.

While there is a well defined discrete generative factor for datasets like MNIST and FashionMNIST, it is less clear what exactly would constitute a discrete factor of variation in datasets like CelebA and Chairs. For example, for CelebA, JointVAE maps various facial identities onto the discrete latent variable. However, facial identity is not necessarily discrete and it is possible that such a factor of variation could also be mapped to a continuous latent variable. JointVAE has a clear advantage in disentangling datasets where discrete factors are prominent (as shown in Fig. 3) but when this is not the case using frameworks that only disentangle continuous factors may be sufficient.

## 4.1 Quantitative evaluation

We quantitatively evaluate our model on the dSprites dataset using the metric recently proposed by Kim & Mnih (2018). Since the dataset is generated from 1 discrete factor (with 3 categories) and 4 continuous factors, we used a model with 6 continuous latent variables and one 3 dimensional discrete latent variable. The results are shown in table 6. Even though the discrete factor in this dataset is not prominent (in the sense that the different categories have very small differences in pixel space) our model is able to achieve scores close to the current best models. Further, as shown in Fig. 6, our model learns meaningful latent representations. In particular, for the discrete factor of variation, JointVAE is able to better separate the classes than other models.

## 4.2 Detecting disentanglement in latent distributions

As noted in Section 2, taken in expectation over data, the KL divergence between the inferred latents $q_\phi(\mathbf{z}, \mathbf{c}|\mathbf{x})$ and the priors, upper bounds the mutual information between the latent units and the data. Motivated by this, we can plot the KL divergence values for each latent unit averaged over a mini batch of data during training. As various factors of variation are discovered in the data, we would expect the KL divergence of the corresponding latent units to increase. This is shown in Fig. 7a. As the capacities $C_z$ and $C_c$ are increased, the model is able to encode more and more factors of variation. For MNIST, the first factor to be discovered is digit type, followed by angle and width. This is likely because encoding digit type results in the largest reconstruction error reduction, followed by encoding angle and width and so on.

After training, we can also measure the KL divergence of each latent unit on test data and rank the latent units by their average KL values. This corresponds to ranking the latent units by how much information they are transmitting about $\mathbf{x}$. Fig. 7b shows the ranked latent units for MNIST and Chairs along with a latent traversal of each unit. As can be seen, the latent units with large information content encode various aspects of the data while latent units with approximately zero KL divergence do not affect the output.

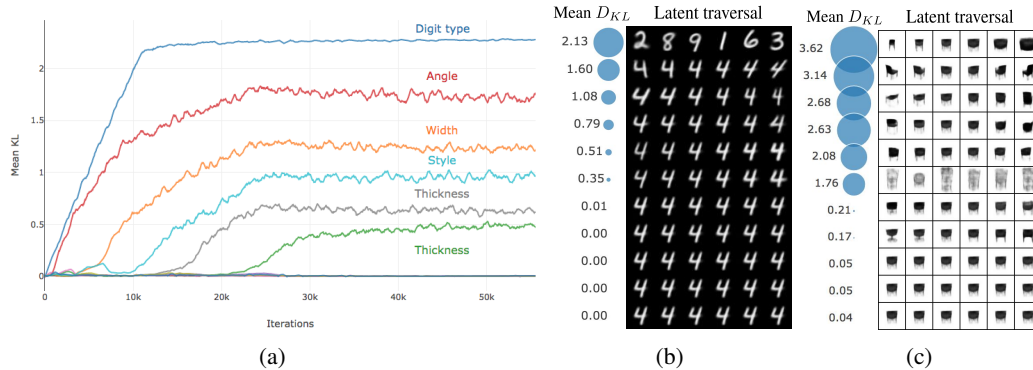

(a)                           (b)                           (c)

Figure 7: (a) Increase of KL divergence during training. As the latent channel capacity is increased, different factors of variation are discovered. Most of the latent units have a KL divergence of approximately zero throughout training, meaning they do not encode any information about the data. As training progresses, however, some latent units start encoding more information about the data. Each latent unit can then be matched to a factor of variation of the data by visual inspection. (b, c) Each row corresponds to a latent traversal of a single latent unit. The column on the left shows the mean KL divergence value over a large number of examples (which corresponds to the amount of information encoded in that latent unit in nats). The rows are ordered from the latent unit with largest KL divergence to the lowest. As can be seen, large KL divergence values correspond to active latents which encode information about the data, whereas low KL divergence value channels to do not affect the data.

## 4.3 The inference network

One of the advantages of JointVAE is that it comes with an inference network $q_\phi(\mathbf{z}, \mathbf{c}|\mathbf{x})$. For example, on MNIST we can infer the digit type on test data with 88.7% accuracy by simply looking at the value of the discrete latent variable $q_\phi(\mathbf{c}|\mathbf{x})$. Of course, this is completely unsupervised and the accuracy could likely be increased dramatically by using some label information.

Since we are learning several generative factors, the inference network can also be used to infer properties which we do not have labels for. For example, the latent unit corresponding to azimuth on the chairs dataset correlates well with the actual azimuth of unseen chairs. After training a model on the chairs dataset and identifying the latent unit corresponding to azimuth, we can test the inference network on images that were not used during training. This is shown in Fig. 8a. As can be seen, the latent unit corresponding to rotation infers the angle of the chair even though no labeled data was given (or available) for this task.

The framework can also be used to perform image editing or manipulation. If we wish to rotate the image of a face, we can encode the face with $q_\phi$, modify the latent corresponding to azimuth and decode the resulting vector with $p_\theta$. Examples of this are shown in Fig. 8b.

## 4.4 Robustness and sensitivity to hyperparameters

While our framework is robust with respect to different architectures and optimizers, it is, like most frameworks for unsupervised disentanglement, fairly sensitive to the choice of hyperparameters (all hyperparameters needed to reproduce the results in this paper are given in the appendix). Even with a good choice of hyperparameters, the quality of disentanglement may vary based on the random seed. In general, it is easy to achieve some degree of disentanglement for a large set of hyperparameters, but achieving complete clean disentanglement (e.g. perfectly separate digit type and other generative factors) can be difficult. It would be interesting to explore more principled approaches for choosing the latent capacities and how to increase them, but we leave this for future work. Further, as mentioned in Section 4, when a discrete generative factor is not present or important, the framework may fail to learn meaningful discrete representations. We have included some failure examples in Fig. 9.

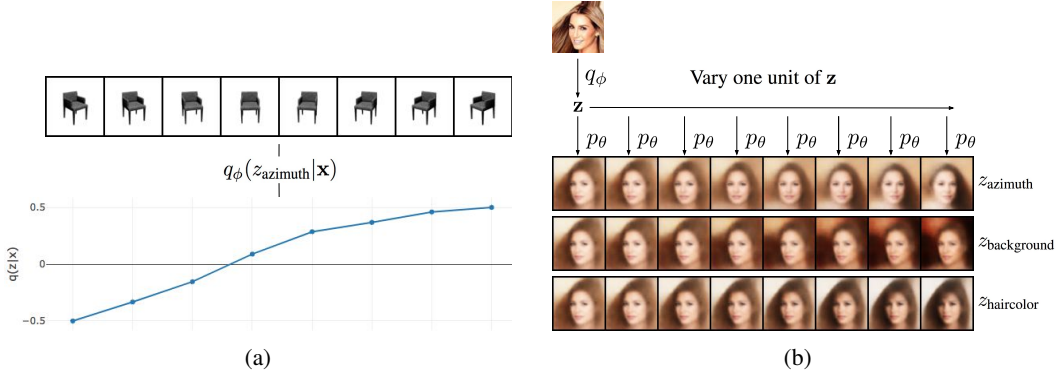

Figure 8: (a) Inference of azimuth on test data of chairs. The first row shows images of chairs from a test set. The second row shows the inferred $z$ for each of the images. As can be seen, the latent unit successfully identifies rotation. (b) Image editing with JointVAE. An image of a celebrity is encoded with $q_\phi$. In the encoded space, we can then rotate the face, change the background color or change the hair style by manipulating the latent unit corresponding to each factor. The bottom rows show the decoded images when each latent factor is changed. The samples are not as sharp as the original image, but these initial results show promise for using disentangled representations to edit images.

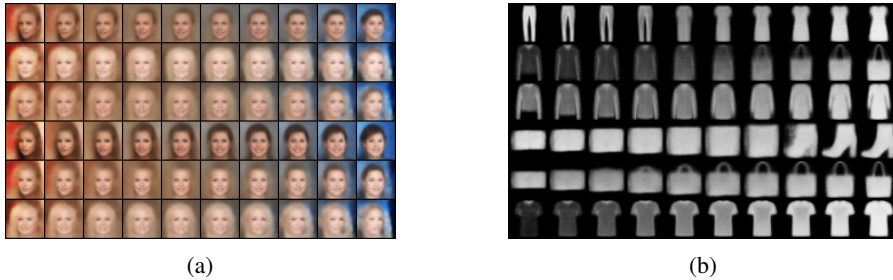

Figure 9: Failure examples. (a) Background color is entangled with azimuth and hair length. (b) Various clothing items are entangled with each other.

## 5    Conclusion

We have proposed JointVAE, a framework for learning disentangled continuous and discrete representations in an unsupervised manner. The framework comes with the advantages of VAEs such as stable training and large sample diversity while being able to model complex jointly continuous and discrete generative factors. We have shown that JointVAE disentangles factors of variation on several datasets while producing realistic samples. In addition, the inference network can be used to infer unlabeled quantities on test data and to edit and manipulate images.

In future work, it would be interesting to combine our approach with recent improvements of the $\beta$-VAE framework, such as FactorVAE (Kim & Mnih (2018)) or $\beta$-TCVAE (Chen et al. (2018)). Gaining a deeper understanding of how disentanglement depends on the latent channel capacities and how they are increased will likely provide insights to build more stable models. Finally, it would also be interesting to explore the use of other latent distributions since the framework allows the use of any joint distribution of reparametrizable random variables.

**Acknowledgments**

The author would like to thank Erik Burton, José Celaya, Suhas Suresha, Vishakh Hegde and the anonymous reviewers for helpful suggestions and comments that helped improve the paper.

## Footnotes

[1] $\beta$-VAE assumes the data is generated by a fixed number of independent factors of variation, so *all* latent variables are in fact conditionally independent. However, for the sake of deriving the JointVAE objective we only require conditional independence between the continuous and discrete latents.

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
