[Supplementary Material]

# Supplementary material

## A  Proofs

### A.1  Expectation of KL divergence and Mutual Information

We can define the joint distribution of the data and the encoding distribution as $q(\mathbf{z}, \mathbf{x}) = p(\mathbf{x})q_\phi(\mathbf{z}|\mathbf{x})$. The distribution of the latent variables is then given by $q(\mathbf{z}) = \mathbb{E}_{p(\mathbf{x})}[q_\phi(\mathbf{z}|\mathbf{x})]$. We can now rewrite the KL divergence between the posterior and the prior taken in expectation over the data as

$$
\begin{aligned}
\mathbb{E}_{p(\mathbf{x})}[D_{KL}(q_\phi(\mathbf{z}|\mathbf{x}) \parallel p(\mathbf{z}))] &= \mathbb{E}_{p(\mathbf{x})}\mathbb{E}_{q_\phi(\mathbf{z}|\mathbf{x})}\left[\log \frac{q_\phi(\mathbf{z}|\mathbf{x})}{p(\mathbf{z})}\right] \\
&= \mathbb{E}_{q(\mathbf{z},\mathbf{x})}\left[\log \frac{q_\phi(\mathbf{z}|\mathbf{x})}{p(\mathbf{z})}\frac{q(\mathbf{z})}{q(\mathbf{z})}\right] \\
&= \mathbb{E}_{q(\mathbf{z},\mathbf{x})}\left[\log \frac{q_\phi(\mathbf{z}|\mathbf{x})}{q(\mathbf{z})}\right] + \mathbb{E}_{q(\mathbf{z},\mathbf{x})}\left[\log \frac{q(\mathbf{z})}{p(\mathbf{z})}\right] \\
&= \mathbb{E}_{q(\mathbf{z},\mathbf{x})}\left[\log \frac{q(\mathbf{z}, \mathbf{x})}{q(\mathbf{z})p(\mathbf{x})}\right] + \mathbb{E}_{q(\mathbf{z})}\left[\log \frac{q(\mathbf{z})}{p(\mathbf{z})}\right] \\
&= I(\mathbf{x};\mathbf{z}) + D_{KL}(q(\mathbf{z}) \parallel p(\mathbf{z})) \\
&\geq I(\mathbf{x};\mathbf{z})
\end{aligned}
$$

where the mutual information is defined under the joint distribution of the data and the encoding distribution.

### A.2  Splitting the discrete and continuous KL divergence terms

Assuming the continuous and discrete latent variables are conditionally independent, i.e. $q_\phi(\mathbf{z}, \mathbf{c}|\mathbf{x}) = q_\phi(\mathbf{z}|\mathbf{x})q_\phi(\mathbf{c}|\mathbf{x})$ and similarly for the prior $p(\mathbf{z}, \mathbf{c}) = p(\mathbf{z})p(\mathbf{c})$ we can rewrite the joint KL divergence as

$$
\begin{aligned}
D_{KL}(q_\phi(\mathbf{z}, \mathbf{c}|\mathbf{x}) \parallel p(\mathbf{z}, \mathbf{c})) &= \mathbb{E}_{q_\phi(\mathbf{z},\mathbf{c}|\mathbf{x})}[\log \frac{q_\phi(\mathbf{z}, \mathbf{c}|\mathbf{x})}{p(\mathbf{z}, \mathbf{c})}] \\
&= \mathbb{E}_{q_\phi(\mathbf{z}|\mathbf{x})}\mathbb{E}_{q_\phi(\mathbf{c}|\mathbf{x})}[\log \frac{q_\phi(\mathbf{z}|\mathbf{x})q_\phi(\mathbf{c}|\mathbf{x})}{p(\mathbf{z})p(\mathbf{c})}] \\
&= \mathbb{E}_{q_\phi(\mathbf{z}|\mathbf{x})}\mathbb{E}_{q_\phi(\mathbf{c}|\mathbf{x})}[\log \frac{q_\phi(\mathbf{z}|\mathbf{x})}{p(\mathbf{z})}] + \mathbb{E}_{q_\phi(\mathbf{z}|\mathbf{x})}\mathbb{E}_{q_\phi(\mathbf{c}|\mathbf{x})}[\log \frac{q_\phi(\mathbf{c}|\mathbf{x})}{p(\mathbf{c})}] \\
&= \mathbb{E}_{q_\phi(\mathbf{z}|\mathbf{x})}[\log \frac{q_\phi(\mathbf{z}|\mathbf{x})}{p(\mathbf{z})}] + \mathbb{E}_{q_\phi(\mathbf{c}|\mathbf{x})}[\log \frac{q_\phi(\mathbf{c}|\mathbf{x})}{p(\mathbf{c})}] \\
&= D_{KL}(q_\phi(\mathbf{z}|\mathbf{x}) \parallel p(\mathbf{z})) + D_{KL}(q_\phi(\mathbf{c}|\mathbf{x}) \parallel p(\mathbf{c}))
\end{aligned}
$$

## B  Model architecture

The architecture of the model is shown in the table. The non linearities in both the encoder and decoder are ReLU except for the output layer of the decoder which is a sigmoid.

For 64 by 64 images (Chairs, CelebA and dSprites) the architecture shown in the table was used. For 32 by 32 images (MNIST and FashionMNIST which were resized from 28 by 28) we used the same architecture with the last conv layer in the encoder and first in the decoder removed.

## C  Training details

Parameters and training details for each model.

| Encoder $q_\phi$ | Decoder $p_\theta$ |
|---|---|
| 32 Conv $4 \times 4$, stride 2 | Linear latent dimension $\times 256$ |
| 32 Conv $4 \times 4$, stride 2 | Linear $256 \times 64 * 4 * 4$ |
| 64 Conv $4 \times 4$, stride 2 | 64 Conv Transpose $4 \times 4$, stride 2 |
| 64 Conv $4 \times 4$, stride 2 | 32 Conv Transpose $4 \times 4$, stride 2 |
| Linear $64 * 4 * 4 \times 256$ | 32 Conv Transpose $4 \times 4$, stride 2 |
| Linear layers for parameters of each distribution | 3 Conv Transpose $4 \times 4$, stride 2 |

## C.1 MNIST

- Latent distribution: 10 continuous, 1 10-dimensional discrete
- Optimizer: Adam with learning rate 5e-4
- Batch size: 64
- Epochs: 100
- $\gamma$: 30
- $C_z$: Increased linearly from 0 to 5 in 25000 iterations
- $C_c$: Increased linearly from 0 to 5 in 25000 iterations

## C.2 FashionMNIST

- Latent distribution: 10 continuous, 1 10-dimensional discrete
- Optimizer: Adam with learning rate 5e-4
- Batch size: 64
- Epochs: 100
- $\gamma$: 100
- $C_z$: Increased linearly from 0 to 5 in 50000 iterations
- $C_c$: Increased linearly from 0 to 10 in 50000 iterations

## C.3 Chairs

- Latent distribution: 32 continuous, 3 binary discrete
- Optimizer: Adam with learning rate 1e-4
- Batch size: 64
- Epochs: 100
- $\gamma$: 300
- $C_z$: Increased linearly from 0 to 30 in 100000 iterations
- $C_c$: Increased linearly from 0 to 5 in 100000 iterations

## C.4 CelebA

- Latent distribution: 32 continuous, 1 10-dimensional discrete
- Optimizer: Adam with learning rate 5e-4
- Batch size: 64
- Epochs: 100
- $\gamma$: 100
- $C_z$: Increased linearly from 0 to 50 in 100000 iterations
- $C_c$: Increased linearly from 0 to 10 in 100000 iterations

### C.5 dSprites

- Latent distribution: 6 continuous, 1 3-dimensional discrete
- Optimizer: Adam with learning rate 5e-4
- Batch size: 64
- Epochs: 30
- $\gamma$: 150
- $C_z$: Increased linearly from 0 to 40 in 300000 iterations
- $C_c$: Increased linearly from 0 to 1.1 in 300000 iterations

Note that since the KL divergence between a categorical variable and a uniform categorical variable is bounded, the discrete capacity is clipped during training if $C_c$ exceeds the maximum capacity. Let $P$ denote a categorical random variable and let $Q$ be a uniform categorical variable, then

$$D_{KL}(P\|Q) = \sum_{i=1}^{n} p_i \log \frac{p_i}{q_i} = \sum_{i=1}^{n} p_i \log \frac{p_i}{1/n} = -H(P) + \log n \leq \log n$$

During training $C_c$ is then clipped as $C_c = \min(C_c, \log n)$.

## D  Things that didn't work

We experimented with several things which we found did not improve disentanglement of joint continuous and discrete representations.

- Modifying the latent distribution in $\beta$-VAE to include a joint Gaussian and Gumbel-Softmax distribution without changing the loss. This generally resulted in the model ignoring the discrete codes.
- Changing the loss function to have a higher $\beta$ on the continuous KL term and a lower $\beta$ on the discrete KL term. For a large combination of $\beta$, we either found the model to ignore the discrete latent codes or to produce representations where continuous factors were encoded in the discrete latent variables.
- In $\beta$-VAE, there is a larger weight on the KL term than in a traditional VAE model. In most VAE models with a Gumbel-Softmax latent variable, the KL divergence between the Gumbel-Softmax variables is approximated by the KL divergence between the corresponding categorical variables. We hypothesized that the approximation error might be worse in $\beta$-VAE, since there is a larger weight on the KL term. Unfortunately, there is no closed form expression of the KL divergence between two Gumbel-Softmax variables. We used various approximations of this, but most estimates had very high variance and impeded learning in the model.

## E  Choice of discrete dimensions

As discussed in the main section of the paper, it is not clear what exactly would constitute a discrete factor of variation for a dataset like CelebA for example. As such, the choice is somewhat arbitrary: when using a 10 dimensional discrete latent variable, the model encodes 10 facial identities and when using more than 10 dimensions it encodes more identities. This was generally found to be quite robust, except when the number of discrete dimensions was exceedingly large (>100), when the model would start to encode e.g. facial angles in the discrete dimensions. Similarly, the reason we choose a 10 dimensional discrete latent variable for MNIST is because we know a priori that there are 10 types of digits. When choosing less than 10 discrete dimensions on MNIST, the model tends to fuse digit types which look similar into one discrete dimension. For example, 4 and 9 or 5 and 8 may correspond to one discrete dimension instead of being separated. When using more than 10 dimensions, the model tends to separate different writing styles of digits into separate dimensions, e.g. 2's with and without a curl at the bottom or 7's with and without a middle stroke may be encoded into different categories.

# F   Comparison with InfoGAN

We include comparisons with InfoGAN which can also disentangle joint continuous and discrete factors. InfoGAN successfully disentangles digit type, from angle and width. However, width and stroke thickness remain entangled. Further, InfoGAN models are typically less stable to train.

(a) Angle (continuous)

(b) Thickness (continuous)

(c) Digit type (discrete)

(d) Width (continuous)

Figure 1: JointVAE.

(a) Angle (continuous)

(b) Thickness and Width (continuous)

(c) Digit type (discrete)

Figure 2: InfoGAN.

# G   Latent traversals

In all figures latent traversals of continuous variables are from $\Phi^{-1}(0.05)$ to $\Phi^{-1}(0.95)$ where $\Phi^{-1}$ is the inverse cdf of a unit normal. Latent traversals of discrete variables are from 1 to the number of dimensions of the variable.