[Reviews · NeurIPS 2018]

Reviewer 1



The paper combines existing ideas of disentangled representation learning (Higgins et al, 2017) and categorical latent variable learning (Maddison et al, 2016) in VAEs to create an architecture capable of learning both continuous and discreet disentangled latent variables. The paper is very well written. It produces compelling results demonstrating disentangling on a variety of datasets. It compares the approach to the relevant baselines and does further analysis to build the readers' intuition. While the novelty is arguably limited (the paper combines two existing approaches in a rather straightforward manner), it is the first demonstration of an unsupervised approach to learning disentangled continuous and discrete latent variables that works well. Hence, I believe the paper has merit to be published at NIPS. Suggestions for improvement: -- I think the paper would be stronger if more discussion was provided around the choice of the number of discrete latent variables. For example, in the MNIST experiment, it would be useful to show how the representation learning is affected by over- or under-specifying the number of discrete dimensions (e.g. <10 or >10). -- It would be good to compare the performance of the model to that of InfoGAN (Chen et al, 2016), since InfoGAN can also learn both discrete and continuous disentangled representations -- How does the model perform when a dataset contains more than 1 discrete latent variable? Is it able to discover that? -- Do the authors have any intuition or metric to measure whether the discrete dimensions are used in a way that is disentangled and improves the model's performance (this is relevant to the discussion point around the Chairs dataset)? -- Why were 10 discrete dimensions used in the CelebA experiment? Presumably there are more than 10 identities in the dataset. Have the authors attempted using >10 discrete dimensions? It would also be useful to see the identities learnt by the model in a plot similar to Fig. 2c. Minor points: -- The reference to Chen et al (2018) on line 15 is misleading, since that paper did in fact address discrete disentangled representation learning -- The Desjardins et al (2012) reference on line 28 is misplaced -- What is the schedule for setting the capacities C_z and C_c? It is worth including these in the main text

Reviewer 2



This paper presents a new framework (JointVAE) for learning disentangled representations.The particular problem that this paper focuses on is the separating discrete and continuous factors of variation in a fully unsupervised manner. Most existing approaches for learning disentangled representations a prior in the form of a Gaussian with diagonal covariance, which is not suitable for capturing categorical features. In the proposed approach a Gaussian distribution represents the continuous features (such as slant and thickness in MNIST) and a Gumbel-Softmax represents categorical features such (as digit-type). Inducing disentanglement is done by penalizing the KL term between the inference distribution and the prior in a controlled capacity manner as proposed by Burgess et al. (2017) which the upper bound on the mutual information between ‘x’ and ‘z’ gradually increases during training. The results shows successful disentanglement on various datasets including MNIST, Fashion-MNIST, and CelebA. This is a well-written paper. The authors do a good job of analyzing both the strengths and weaknesses of their proposed method for disentangling discrete features in different datasets. I think that the main of the paper lies in the relatively thorough experimentation. I thought the results in Figure 6 were particularly interesting in that they suggest that there is an ordering in features in terms of mutual information between data and latent variable (for which the KL is an upper bound), where higher mutual information features appear first as the capacity is increased. I also appreciate the explicit discussion of the robust of the degree of disentanglement across restarts, as well as the sensitivity to hyperparameters. Given the difficulties observed in Figure 4 in distinguishing between similar digits (such as 5s and 8s), it would be interesting to see results for this method on a dataset like dSprites, where the shapes are very similar in pixel space. The inferred chair rotations in Figure 7 are also a nice illustration of the ability of the method to generalize to the test set. The main thing that this paper lacks is a more quantitative evaluation. A number of recent papers have proposed metrics for evaluating disentangled representations. In addition the metrics proposed by Kim & Mnih (2018) and Chen et al. (2018), the work by Eastwood & Williams (2017) [1] is relevant in this context. All of these metrics presume that we have access to labels for true latent factors, which is not the case for any of the datasets considered in the experimentation. However, it would probably be worth evaluating one or more of these metrics on a dataset such as dSprites. A minor criticism is that details the training procedure and network architectures are somewhat scarce in the main text. It would be helpful to briefly describe the architectures and training setup in a bit more detail, and explicitly call out the relevant sections of the supplementary material. In particular, it would be good to list key parameters such as γ and the schedule for the capacities Cz and Cc, e.g., the figure captions. In Figure 6a, please mark the 25k iterations (e.g. with a vertical dashed line) to indicate that this is where the capacity is no longer increased further. Questions - How robust is the ordering on features Figure 6, given the noted variability across restarts in Section 4.3? I would hypothesize that the discrete variable always emerges first (given that this variable is in some sense given a “greater” capacity than individual dimensions in the continuous variables). Is the ordering on the continuous variables always the same? What happens when you keep increasing the capacity beyond 25k iterations. Does the network eventually use all of the dimensions of the latent variables? - I would also appreciate some discussion of how the hyperparameters in the objective were chosen. In particular, one could imagine that the relative magnitude of Cc and Cz would matter, as well as γ. This means that there are more parameter to tune than in, e.g., a vanilla β-VAE. Can the authors comment on how they chose the reported values, and perhaps discuss the sensitivity to these particular hyperparameters in more detail? - In Figure 2, what is the range of values over which traversal is performed? Related Work In addition to the work by Eastwood & Williams, there are a couple of related references that the authors should probably cite: - Kumar et. al [2] also proposed the total correlation term along with Kim & Mnih (2018) and Chen et al. (2018). - A recent paper by Esmaeli et al. [3] employs an objective based on the Total Correlation, related to the one in Kim & Mnih (2018) and Chen et. al (2018) to induce disentangled representations that can incorporate both discrete and continuous variables. Minor Comments - As the authors write in the introduction, one of the purported advantages of VAEs over GANs is stability of training. However, as mentioned by the author, including multiple variables of different types also makes the representation unstable. Given this observation, maybe it is worth qualifying these statements in the introduction. - I would say that section 3.2 can be eliminated - I think that at this point readers can be presumed to know about the Gumbel-Softmax/Concrete distribution. - Figure 1 could be optimized to use less whitespace. - I would recommend to replace instances of (\citet{key}) with \citep{key}. References [1] Eastwood, C. & Williams, C. K. I. A Framework for the Quantitative Evaluation of Disentangled Representations. (2018). [2] Kumar, A., Sattigeri, P. & Balakrishnan, A. Variational inference of disentangled latent concepts from unlabeled observations. arXiv preprint arXiv:1711.00848 (2017). [3] Esmaeili, B. et al. Structured Disentangled Representations. arXiv:1804.02086 [cs, stat] (2018).

Reviewer 3



Update following rebuttal: I thank the authors for their thoughtful reply to my questions. I maintain my assessment that this is a a very good paper. ----- The authors present an interesting exploration of combining joint and discrete distributions into a fully factorized latent representation in the context of a beta-VAE framework. The concept itself is only moderately novel, as all of the components (the beta-VAE framework, the forced capacity technique, Gumbel-softmax distributions, etc.) are based on previous work, and the authors merely combine them, but the results are nevertheless convincing and interesting. It's good to know that assuming an appropriate prior for the data (e.g., a discrete variable with 10 dimensions for MNIST combined with a set of continuous degrees of freedom) actually helps in disentangling the data distribution, compared with assuming a strictly Gaussian prior. One question that I would like to see addressed, which the authors apparently missed is: is the forced capacity technique (i.e. annealing the values of C_z and C_c during training) still necessary to achieve disentanglement when the joint prior is used, or could the same level of disentanglement be achieved without it, just by virtue of having a more appropriate model for the data? A minor improvement to the paper would be to give Figure 4a its own number, as it is a bit confusing to have it in the same figure as 4b and 4c, which show a different dataset.